# Domestication over Speciation in Allopolyploid Cotton Species: A Stronger Transcriptomic Pull

**DOI:** 10.3390/genes14061301

**Published:** 2023-06-20

**Authors:** Josef J. Jareczek, Corrinne E. Grover, Guanjing Hu, Xianpeng Xiong, Mark A. Arick II, Daniel G. Peterson, Jonathan F. Wendel

**Affiliations:** 1Ecology, Evolution, and Organismal Biology Department, Iowa State University, Ames, IA 50010, USA; jjareczek01@bellarmine.edu (J.J.J.); corrinne@iastate.edu (C.E.G.); 2Biology Department, Bellarmine University, Louisville, KY 40205, USA; 3State Key Laboratory of Cotton Biology, Institute of Cotton Research, Chinese Academy of Agricultural Sciences, Anyang 455000, China; huguanjing@caas.cn; 4Shenzhen Branch, Guangdong Laboratory of Lingnan Modern Agriculture, Key Laboratory of Synthetic Biology, Ministry of Agriculture and Rural Affairs, Agricultural Genomics Institute at Shenzhen, Chinese Academy of Agricultural Sciences, Shenzhen 518120, China; xiongxianpeng@caas.cn; 5Institute for Genomics, Biocomputing & Biotechnology, Mississippi State University, Mississippi State, MS 39762, USA; maa146@igbb.msstate.edu (M.A.A.II); peterson@igbb.msstate.edu (D.G.P.)

**Keywords:** cotton, cotton fiber, domestication, *Gossypium hirsutum*, *Gossypium barbadense*, fiber development

## Abstract

Cotton has been domesticated independently four times for its fiber, but the genomic targets of selection during each domestication event are mostly unknown. Comparative analysis of the transcriptome during cotton fiber development in wild and cultivated materials holds promise for revealing how independent domestications led to the superficially similar modern cotton fiber phenotype in upland (*G. hirsutum*) and Pima (*G. barbadense*) cotton cultivars. Here we examined the fiber transcriptomes of both wild and domesticated *G. hirsutum* and *G. barbadense* to compare the effects of speciation versus domestication, performing differential gene expression analysis and coexpression network analysis at four developmental timepoints (5, 10, 15, or 20 days after flowering) spanning primary and secondary wall synthesis. These analyses revealed extensive differential expression between species, timepoints, domestication states, and particularly the intersection of domestication and species. Differential expression was higher when comparing domesticated accessions of the two species than between the wild, indicating that domestication had a greater impact on the transcriptome than speciation. Network analysis showed significant interspecific differences in coexpression network topology, module membership, and connectivity. Despite these differences, some modules or module functions were subject to parallel domestication in both species. Taken together, these results indicate that independent domestication led *G. hirsutum* and *G. barbadense* down unique pathways but that it also leveraged similar modules of coexpression to arrive at similar domesticated phenotypes.

## 1. Introduction

Domestication of wild plants and animals has been a prominent feature of human history for thousands of years, leading to the expansion of civilizations worldwide [1]. Many plant and animal species have been domesticated by humans, from food, labor, and companion animals to the crop species that make up the backbone of modern agriculture [1,2,3]. Crops, in particular, have been instrumental in human history, with cereal grains such as barley, wheat, and rice kickstarting the transition to an agricultural society. When viewed through a scientific lens, domestication provides an opportunity to explore evolution on a telescoped timescale. The strong directional selection that species undergo during domestication causes phenotypic changes within thousands of years rather than millions [2,4,5,6,7,8,9]. Examining the impact of domestication on genomes, transcriptomes, proteomes, and the other -omes can illuminate how they have responded to human selection, which in turn provides insight into how these processes might operate in natural settings for non-domesticated plants and animals.

A common observation is that domesticated plants exhibit similar phenotypic responses to similar selection pressures [9,10,11]. Fruit size and yield tend to increase, plant size and architecture become modified to permit compact planting, and plants acquire day-length neutrality in flowering to facilitate broader habitats and synchronized harvest, as seen in maize, cotton, rice, and dozens of other plant species [4,5,12,13,14,15]. Domestication is not without its problematic consequences [16], however, as domesticated plants tend to be more susceptible to diseases and pests, have lower genetic diversity than their wild ancestors due to bottlenecks of the domestication process, and require considerable agricultural inputs. These recurring themes in domesticated plants are of great interest, particularly in plants where closely related taxa have been independently domesticated and, therefore, potentially arrived at similar phenotypes via some mix of parallel and independent genetic changes.

One genus that provides an excellent system to study multiple domestication events is *Gossypium*, the source of cotton and the most prevalent textile plant in the world [17]. Agronomically, cotton refers to one of four species that were independently domesticated on two continents roughly 5–8 thousand years ago. Notably, the two species that dominate cotton commerce, *G. hirsutum* (AD_1_) and *G. barbadense* (AD_2_), are allopolyploid, like many crops, meaning they originally had duplicated copies of every gene. *Gossypium hirsutum*, or Upland cotton, is the most widely grown species, accounting for over 90% of commercially grown cotton globally [17]. Native populations of *G. hirsutum* are spread across Central America and the Caribbean; the species was most likely domesticated initially in the northern part of the Yucatan Peninsula, from where it spread under domestication throughout Central America, the Caribbean, and much later, the US and globally [18]. *Gossypium barbadense*, also known as Egyptian cotton or Sea Island cotton, makes up roughly 8% of commercially grown fiber from the two allopolyploid cotton species [17]. Native to and domesticated within coastal Peru west of the Andes, *G. barbadense* expanded into northern South America, Central America, the Caribbean and the Pacific [18,19,20,21]. Several features distinguish the two allopolyploids, with *G. hirsutum* able to tolerate a wider range of habitats and with a higher yield, and the two species differing in pest and disease severities. *Gossypium barbadense* has longer, stronger, and finer fiber and is therefore grown more for luxury textiles (colloquially known as “Pima” or “Egyptian” cotton) despite its lower yield. As the world’s foremost textile crops, these two species are the subject of considerable study and efforts at crop improvement, including intentional and unintentional interspecific hybridization. This introgression has occurred many times in both directions, with the goal of producing varieties that possess the positive traits of both *G. hirsutum* and *G. barbadense*, i.e., are relatively hardy and easy to cultivate, and are high-yielding with long, strong, fine fiber [18,20,22,23,24,25].

Despite the importance of cotton to the global economy, the genomic targets of selection and their effects on the domesticated cotton phenotype (day length neutrality, longer, stronger fiber, higher yield, etc.) are mostly unknown. Low genetic diversity is present in wild and domesticated accessions of both species [18,20,22], and the genome-wide targets of domestication are not well understood [18,26]. Comparative transcriptomics has the potential to reveal the targets of selection without the necessity of large-scale GWAS studies or selection screens, thus forming a complementary approach for understanding the mechanisms of domestication. Previously, the cotton fiber transcriptome was evaluated at key timepoints between wild and domesticated *G. hirsutum* [27] or between cultivars of *G. hirsutum* and *G. barbadense* [28,29]. Other analyses have been used to evaluate key genes and/or modules in cotton fiber development between different domesticated accessions or domesticated-derived mutants [30,31,32,33,34,35,36,37,38,39,40]. This study examines the transcriptome of developing fiber in *G. barbadense* at important developmental time points that mirror those previously analyzed [27]. Comparisons were made between these time points, domestication states, and *G. hirsutum* using differential gene expression analysis. Because genes operate in the context of entire networks of genes, we also employed network analysis, allowing insight into how domestication has altered the fiber transcriptome on several levels. This approach offers the opportunity to explore whether selection has differentially operated on various components of the fiber coexpression network in each species and under parallel domestication.

This study addresses two primary questions: (1) How has the transcriptome been altered by the natural process of speciation, playing out over almost a million years, versus domestication, on the timescale of 5000–8000 years? (2) How have the coexpression networks of these two species changed over time in terms of module preservation, membership, and connectivity? Our analyses supported several novel conclusions. First, *G. barbadense* has lower differential gene expression than *G. hirsutum* and fewer significant modules, potentially indicating a more canalized development program. Second, gene expression differences between domesticated types of *G. hirsutum* and *G. barbadense* are greater than between wild forms of the two species, indicating that selection increased expression divergence, even with interspecific gene flow. Third, whereas most coexpression network modules differ between species, there is evidence of parallel selection between the two species. Fourth, modules are not well-preserved between species, indicating a high level of divergence in fiber coexpression networks, which was also increased by domestication. Our results reveal the extraordinary divergence in the transcriptomic underpinnings of the generally similar phenotypes of two independently domesticated cotton species.

## 2. Materials and Methods

### 2.1. Plant Material, RNA Extraction and Sequencing

The developing fiber was collected from *G. hirsutum* and *G. barbadense* plants grown in the Pohl Conservatory at Iowa State University (Table 1). Plants were grown in 2-gallon pots between 22.2–25.5 °C. Flowers were hand-pollinated and tagged, and bolls were collected at 5, 10, 15, and 20 days post-anthesis (DPA) per species (Appendix A). A minimum of three biological replicates were collected for each species at each DPA, where replicates were derived from three different accessions for each combination of species and domestication status (Table 1), thereby reducing the influence of accession-specific expression patterns on each comparison. Bolls were pooled at 5 DPA to obtain sufficient material for RNA extraction. Collected ovules were flash-frozen in liquid nitrogen and stored at −80 °C until RNA extraction. The frozen fibers were first ruptured with glass beads and then subjected to RNA extraction using the Sigma Plant Spectrum total RNA kit (Sigma-Aldrich Burlington, Burlington, MA, USA). The extracted RNA samples were further purified using the phenol-chloroform method as previously described [41]. Samples were sent to the Iowa State University DNA facility for quality control (utilizing the Agilent 2100 Bioanalyzer), library construction, and sequencing on the Illumina NovaSeq 6000 to obtain paired-end (PE), 150 nucleotide reads. Raw reads were quality trimmed using trimmomatic version 0.39 [42] from Spack [43] (module trimmomatic/0.39-da5npsr). Specifically, adapter sequences were removed, and reads were quality trimmed to a minimum length of 75 bp; only reads surviving as PE were kept.

### 2.2. Reference Preparation, Mapping, and Differential Gene Expression Analysis

Species-specific reference transcriptomes were generated using the *G. raimondii* [44] genome annotation and species-specific SNP information [45] using custom scripts from https://github.com/Wendellab/AD2-vs-AD1 (accessed on 16 June 2023). Read mapping and quantification were performed for each sample relative to its representative transcriptome via Kallisto v0.46.1 [46] using ‘kallisto quant’. Samples with fewer than 1 million (M) reads quantified were excluded from the following data analyses conducted in R/4.2.0 [47]. Using the R package DESeq2 v1.36.0 [48], raw read counts were normalized by applying a variance stabilizing transformation (*rld*) with the design ‘~species+condition+DPA’, followed by principal component analysis (PCA) via *plot PCA*. Samples with irregular placement by PCA, potentially representing pre-aborted bolls or mistagged samples, were removed and noted in Appendix A.

Analyses of differential gene expression (DGE) between species, conditions (representing domestication status), and timepoints were conducted using the simplified design “~group”, where the single factor “group” represented all 16 combinations of the three original factors of “species’’ (*G. hirsutum* or *G. barbadense*), “condition” (wild or domesticated), and “DPA” (5, 10, 15, or 20 DPA). Pairwise comparisons were conducted as contrasts between adjacent timepoints in wild and domesticated accessions of the same species (e.g., 10 versus 5 DPA in wild *G. barbadense*), between wild and domesticated accessions of the same species at a given timepoint (e.g., domesticated versus wild *G. barbadense* at 5 DPA), and between species at a given timepoint and for a given condition (e.g., wild *G. barbadense* versus wild *G. hirsutum* at 5 DPA). In all cases, differential expression was considered significant at a Benjamini-Hochberg [49] adjusted *p*-value < 0.05. Overlap between species and timepoints was visualized as UpSet diagrams in R using ComplexUpset v1.3.3 [50] and ggplot2 v3.3.6 [51]. Data tables were generated using tidyverse v1.3.1 [52], magrittr v2.0.3 [53], data.table v1.14.2 [54], and DEGreport v1.32.0 [55]. Relevant code is available from https://github.com/Wendellab/AD2-vs-AD1 (accessed on 16 June 2023).

### 2.3. Weighted Co-Expression Gene Network Analysis

Network analysis was conducted in R using WGCNA [56,57] to build independent weighted gene coexpression networks for *G. hirsutum* and *G. barbadense*, as well as construct a meta-network from all samples, as previously described [58]. Briefly, the raw read count data were filtered to remove genes with zero variance and then *rld* normalized as input to construct a Pearson correlation matrix between all gene pairs. An adjacency matrix was generated from the Pearson correlation matrix to represent gene connection strength. The sum of connection strengths for a gene represents the connectivity of a gene, which indicates how strongly that gene is coexpressed with the other genes in the network. The adjacency matrix was also used to calculate a topological overlap matrix, which measures the strength of coexpression relationships between any two genes with respect to the rest of the network [59]. Genes with highly similar coexpression patterns were clustered into modules [56]. GO enrichment of modules was performed using topGO [60]. Network preservation was measured using *Zsummary* and *medianRank* [56] scores built-in WGCNA.

## 3. Results

### 3.1. RNA-Seq Sample Quality and Removal

From the 80 *G. hirsutum* (AD_1_) samples collected, 39 were removed due to poor read depth, incomplete replication, or unusual placement in the principal component analysis (PCA), the latter perhaps attributable to the occurrence of frequent boll abortion and/or pest damage on *G. hirsutum*. After sample cleaning, 41 *G. hirsutum* samples remained with an average of 27 million mapped reads (Appendix A). Similarly, 18 of the 47 *G. barbadense* (AD_2_) samples were removed due to poor read depth, incomplete replication, or unusual placement in the PCA due to the aforementioned boll abortion and/or pest damage. After sample cleaning, 29 *G. barbadense* samples were retained, with an average of 15.8 million mapped reads. Full sampling information is in Appendix A.

PCA of expression data for the two species (Appendix A) generally separates developmental timepoints on the first axis (33% of the variance) and species on the second axis (12% variance). As previously observed for *G. hirsutum* [27] and reiterated here, *G. barbadense* also displays a general developmental gradient (Appendix A), with early developmental timepoints (i.e., 5 and 10 DPA) generally clustering together and 15 DPA forming a bridge between these and later developmental timepoints (20 and 25 DPA; 25 DPA not displayed). In comparison with timepoint and species, domestication status explains less variance. When examining the single-species PCA results, separate clusters of wild and domesticated samples were evident, especially at the later developmental timepoints of 15 and 20 DPA (Appendix A). Congruent with their status as both independent species and independent domesticates, no interspecific domestication-based clusters are observed.

### 3.2. Differential Expression of Convergent Domesticates and Their Wild Progenitors

Differential gene expression (DGE) analyses between species, with respect to domestication status and among time points, are summarized in Figure 1 and Table 2. Generally, the extent of differential expression in *G. hirsutum* is greater than in *G. barbadense* for any given comparison (between timepoints and/or wild versus domesticated). This difference could be due in part to the lack of truly wild accessions of *G. barbadense*, to a more uniform developmental program in *G. barbadense*, and/or perhaps because a larger number of morphogenetic transformations were selected in *G. hirsutum* than in *G. barbadense*. In *G. barbadense*, the greatest amount of DGE occurs between 15 and 20 DPA in both the wild and domesticated accessions, whereas DGE between the previous timepoints was substantially lower. Likewise, the later DPA timepoints (i.e., 15 and 20 DPA) exhibited greater DGE between wild and domesticated *G. barbadense* than the earlier timepoints (i.e., 5 and 10 DPA). Similar patterns were both previously observed in *G. hirsutum* [27] and reiterated here using additional sampling; in *G. hirsutum*, however, the number of DGE between 5 and 10 DPA is more similar to the number of DGE observed between 15 and 20 DPA, in contrast to *G. barbadense*, in which substantial DE was only observed between 15 and 20 DPA. Interestingly, for both *G. hirsutum* and *G. barbadense*, DGE between timepoints is generally higher in the wild accessions than in domesticates, except for 10 to 15 DPA in both species, possibly indicating a difference in development timing relative to chronological DPA. Also notable is the amount of DGE between wild and domesticated *G. hirsutum*, which far exceeds the DGE between wild and domesticated in *G. barbadense* (24,627 total DEG in *G. hirsutum* vs. 7608 in *G. barbadense*).

Interspecific comparisons between *G. hirsutum* and *G. barbadense* reveal substantial DGE for both wild and domesticated comparisons (e.g., *G. hirsutum* wild versus *G. barbadense* wild) at all timepoints, for a total of 33,295 differentially expressed genes in the wild and 58,805 in the domesticated (Table 2, “interspecies’’ column). This is notable as a dramatic difference in the transcriptomic usage (45% versus 79% of all genes) in the same structure of two closely related species. Interestingly, the amount of DGE was greater for the domesticated interspecific comparisons than for the wild interspecific comparisons on a per DPA basis. In other words, the fiber expression profiles of *G. hirsutum* and *G. barbadense* became more divergent after domestication despite intense directional selection for similar characteristics. These interspecific differences in expression are reiterated in the general lack of shared DGE genes in any comparison (Figure 2); however, there are between 5 and 96 genes that exhibit similar novel patterns of DE between DPA in the domesticated accessions of both species (Figure 2, the intersection between AD_1_ and AD_2_ domesticated sets; Appendix A). These genes include several related to reactive oxygen species management, an important aspect of fiber elongation, upregulated between 10 and 15 DPA: peroxidase (Gorai.001G001800.A, Gorai.008G080800.D) and oxidoreductase genes (Gorai.002G062200.A, Gorai.008G296500.A, Gorai.013G179000.A). A xyloglucan endotransglucosylase/hydrolase gene involved in cell wall polysaccharide metabolism is also upregulated between 10 and 15 DPA (Gorai.005G153200.A), as is a gene related to the cytoskeleton, crucial at all stages of fiber development (Gorai.005G168500.A). Interestingly, while no gene exhibited similar DE in *G. hirsutum* and *G. barbadense* for more than one developmental timepoint, eight homoeologous pairs appeared on this list of putative convergent DE genes (i.e., homoeologous pairs for Gorai.001G218000, Gorai.004G062100, Gorai.005G042100, Gorai.009G215100, Gorai.009G244800, Gorai.010G194600, Gorai.010G202700, and Gorai.011G182600).

In all but one instance, the two homoeologs exhibited DGE at the same timepoint and in the same direction, with the noted exception (i.e., Gorai.004G062100; “ethylene forming enzyme”) exhibiting upregulation of the D-homoeolog early in development and the A-homoeolog exhibiting upregulation later in development (Appendix A). Overall, the number of homoeologs that exhibited similar expression differences between domesticated *G. hirsutum* and *G. barbadense* was similar for both subgenomes (151 A-homoeologs versus 146 D).

Because only one of the polyploid parents produces spinnable fiber (i.e., the “A-genome”), we also directly compared global homoeolog expression bias by evaluating DE between homoeologs. Contrary to expectations, more homoeolog pairs exhibited a general D-genome bias among genes in 14 out of 16 samples (Test of Proportions, *p* < 0.05; Appendix A). Notably, comparisons between conditions (i.e., wild versus domesticated for the same species and DPA) suggest only one significant difference (Test of Proportions, *p* < 0.05), albeit in the opposite direction than expected. The only inter-condition difference detected was between wild and domesticated *G. barbadense* at 10 DPA, where the domesticated form exhibited a greater bias toward the D-genome.

### 3.3. Meta-Coexpression Network Analysis Fiber Development in the Two Allopolyploid Species

To provide a global view of cotton fiber gene co-expression relationships, a meta-coexpression network analysis was conducted using all 70 samples (41 from *G. hirsutum* and 29 from *G. barbadense*). After removing invariant genes and those with zero expression, a network of 69,686 genes was constructed comprising 58 coexpression modules (Appendix A). Applying a multi-factor design to examine the representative module co-expression profiles (*eigengene ~ species + development + domestication*), most (56) of these modules exhibited significant associations (using ANOVA) with species (35), domestication status (40), and/or developmental timepoints (39). Applying a simplified design that combined the original three factors into a single factor (*species_domestication_development*), nearly all modules (53 out of 58) also exhibited significant associations. These results serve not just to reinforce the genome-wide transcriptomic alterations that have accompanied speciation and domestication but also reveal a network or “modular” view of these changes. Functional enrichment (Appendix A) using topGO revealed nine modules with clear relevance to fiber development, as implied by the gene ontology of module member genes (Appendix A). Representative expression patterns of these modules are shown in Figure 3.

Two of these fiber-relevant modules, ME3 and ME8, are functionally enriched for polysaccharide synthesis and transport, which are important to developing fiber and the high cellulose content of cotton fiber [61]. ME3 exhibits a gradual upregulation of module member genes from 5 to 20 DPA in all four accessions studied, with more drastic changes in *G. hirsutum* than in *G. barbadense*. ME3 contains ~8300 genes (Appendix A), among which are expansin, sucrose synthase, cellulose synthase, pectin lyase, actin-related protein, actin depolymerizing factor, tubulin, myosin, and profilin, all of which are genes that play a role in cell wall synthesis, particularly secondary cell wall synthesis, which is a critical aspect of fiber development [62,63,64,65,66,67,68,69]. Among those are four negative regulators of fiber length (*GhMYB1*, *GhKNL1*, *GhFSN1*, *GhBZR3*), whose higher expression in *G. hirsutum* (versus *G. barbadense*) likely contributes to the shorter fibers in upland cotton (Appendix A). ME8, which has significant associations with domestication and development, has a less clear expression pattern; it is generally more highly expressed at 10 and 15 DPA and in wild accessions. ME8 contains roughly 2000 genes (Appendix A), some of which are relevant to fiber development, namely pectin lyase, pectin methylesterase, cellulose synthase-like protein, sucrose synthase, expansin, and callose synthase, all of which are related to cell wall development [62,70,71,72].

Three of the nine modules (ME2, ME3, and ME7) are associated with reactive oxygen species physiology, a process that is important for cell function but also plays a role in fiber development by impacting cell wall extensibility and signaling the onset of secondary cell wall synthesis [67,73,74,75]. ME2 has significant associations with domestication, species, and development and is more highly expressed in *G. barbadense* than in *G. hirsutum*. Two ME2 member genes encoding the rate-limiting enzymes GhACO1 and GhACO2 in ethylene biosynthesis are known to play a positive regulatory role in fiber development (Appendix A). ME7 has significant associations for both species and domestication. It is most highly expressed in domesticated *G. hirsutum*, with much lower expression in all other accessions and at most other timepoints. ME7 contains ~2600 genes (Appendix A) with a wide variety of functional annotations; some that are relevant to fiber development include methyltransferases, myosins, expansin, oxidoreductases, peroxidases, α tubulin, formins, and MYB-domain proteins [34,63,68,76,77,78].

One module, ME6, was related to RNA synthesis and cell cycle regulation, the latter possibly relevant to the fiber suppressing cell division. For example, a Gibberellic Acid (GA) receptor gene *GhGID1a*, which is involved in the regulation of the GA signaling pathway, was found in ME6, and its ectopic expression in *Arabidopsis* leads to reduced growth (Appendix A). The coexpression profile of ME6 exhibited significant associations with all three terms (i.e., species, domestication, and development) and is generally more highly expressed in wild *G. hirsutum*. It also exhibits low expression at 5 DPA and generally higher expression at later DPA in all species. Two modules (ME7 and ME14) were associated with cytoskeletal development, motor proteins, and intracellular transport. These functions are all crucial in developing fibers, as the cytoskeleton plays a role in cell shape and in transporting materials to the site of growth in a cell. ME7, discussed above, is notable because it is only highly expressed in domesticated *G. hirsutum*, and ME14 had significant associations with domestication. This module is most highly expressed in domesticated *G. hirsutum*, with high expression at later timepoints in domesticated *G. barbadense* as well, thus comprising a second example of a module that may reflect parallel change under dual domestication. ME14, which contains 839 genes (Appendix A), exhibits low expression in wild accessions of both species and contains several genes relevant to cellular transport, including vesicle-associated proteins, SNARE proteins, actin, profilin, dynein, tubulin, and clathrin adaptor complex proteins [70,78,79,80,81]. One module, ME16, is related to autophagy, a critical cellular process that likely plays a role in fiber maturation. Interestingly, ME16 significantly correlates with all three factors (i.e., species, domestication, and development) and displays high expression levels at 20 DPA. ME24 is related to nutrient reservoir activity; sucrose management is a critical part of cellulose synthesis, which is, in turn, an essential aspect of fiber development. ME24 is significant under domestication, species, and development, shows high expression at 20 DPA in both species and domestication states, and contains genes related to polysaccharide synthesis and transport (Appendix A). The final module relevant to fiber development, ME34, is related to actin-based transport and is associated with domestication state and development. ME34 exhibits higher expression in domesticated accessions of both species but has different expression patterns between *G. hirsutum* and *G. barbadense*, perhaps indicating a target of parallel selection. It has high expression at 5 and 20 DPA in *G. hirsutum* but a more even pattern expression through development in *G. barbadense*. It contains several actin-related genes (Appendix A), such as villin, decapping protein, myosin, kinesin, and formin [63,66,79,82].

Two modules that did not have significant associations under domestication nevertheless displayed notable patterns of expression over time or between species. Both ME4 (ROS management) and ME5 (DNA binding) showed strong species and development effects, with strong eigengene patterns (Appendix A). Specifically, ME4 was more highly expressed in *G. hirsutum* throughout development than in *G. barbadense*, while the reverse was true for ME5. These modules, containing 3583 and 3080 genes (Appendix A), respectively, may represent fundamental species expression differences that remain constant across the domestication divide in both species.

### 3.4. Comparison between the Separate Species-Networks for G. hirsutum and G. barbadense

In addition to the meta-network analysis constructed for both species, species-specific networks of fiber domestication were generated for interspecific comparison of the underlying transcriptional organization between *G. hirsutum* and *G. barbadense*. The *G. hirsutum* network comprised 62,087 genes partitioned into 76 modules (Appendix A). About one-half of these modules (35) demonstrated significant associations with *domestication* (23) or *development* (23), while 31 had significant associations when tested for the combined factor of *domestication_development* (Appendix A). Functional enrichment (Appendix A) of these modules revealed six modules with clear relevance to fiber development (Figure 4a, Appendix A). ME3 is significantly associated with *development* and *domestication_development* and is related to the microtubule cytoskeleton and cellular transport. It is more highly expressed in wild accessions of *G. hirsutum* and tends to have high expression at 10 and 15 DPA. ME3 contains 5096 genes (Appendix A); some genes of interest include the annexin gene GbAnx6 (Appendix A; [83]) and others encoding tubulin, dynamin, microtubule-associated proteins, actin-related proteins, and several types of transferase [63,79]. ME14 and ME33 are also related to cellular transport and are significantly associated with *development* and *domestication_development*, and ME33 is additionally associated with *domestication*. ME14 displays higher expression at 5 and 20 DPA in both wild and domesticated accessions, and ME33 has high expression at 5 DPA and generally high expression in wild accessions. ME14 has ~1300 genes (Appendix A), including dynamin, myosin, actin depolymerizing factor, actin-related proteins, formin, and villin [63,64,66,79,82]. Among those, GhFLA1 is a fasciclin-like arabinogalactan protein involved in the Cdc42p-dependent organization of the actin cytoskeleton [84] that has been reported to positively regulate fiber elongation (Appendix A). ME5 is related to reactive oxygen species management, is significant for *domestication*, *development*, and *domestication_development*, and its expression is inversely correlated with DPA, with a sharp drop in expression between 5 and 10 DPA. ME5 contains ~4600 genes (Appendix A), including several oxidoreductases and peroxidases [75,85]. In addition, the transcription factor *GhbHLH18* was also present in ME5, which regulates fiber development through the activation of peroxidase-mediated lignin metabolism [86]. Another five known functional genes in ME5 (i.e., GhSMT2–1, GhLTPG1, GhHOX3, GhGalT1 and GhACT_LI1) may play an important role in the domestication of cotton fibers [87,88,89,90,91]. ME12 is related to autophagy and is associated with *domestication* and *domestication_development*. It shows much higher expression in domesticated *G. hirsutum* than in wild *G. hirsutum*. The final module, ME48, is related to polysaccharide synthesis. It is significant for *domestication* and *domestication_development*, and it shows higher expression in wild *G. hirsutum* than in domesticated *G. hirsutum*. ME48 contains 148 genes (Appendix A), including those encoding several hydrolases, cellulose synthase, and cellulose synthase-like protein [71,92,93]; however, none have been previously associated with fiber development.

The *G. barbadense* network consisted of 61,934 genes (Appendix A) partitioned into 56 modules, 20 fewer than were detected in *G. hirsutum*. Of these 56 modules, 21 had significant associations with *domestication* (12), *development* (13), and/or *domestication_development* (19). Functional enrichment of these modules revealed 6 with clear relevance to fiber development (Figure 4b, Appendix A). ME3 is related to polysaccharide synthesis and is significantly associated with *domestication*, *development*, and *domestication_development*. Expression of ME3 is correlated with DPA, with a sharp increase in expression between 15 and 20 DPA. ME3 contains ~4500 genes (Appendix A), including those encoding the following proteins of interest: cellulose synthase and cellulose synthase-like protein, galactosyltransferase, sucrose synthase, exostosin, glycosyl transferase, callose synthase, and several hydrolases [62,71,92,94,95]. Fourteen known functional genes, including a sucrose synthase gene (GhSusA1) and two cellulose synthase genes (*GhcelA1* and *GhcelA2*), were present in ME3, indicating the significance of this module in fiber development (Appendix A). ME4 is related to the actin cytoskeleton and is significantly associated with all three test conditions. In domesticated *G. barbadense*, this module displays expression correlated with DPA, with a sharp increase between 10 and 15 DPA that continues upward into 20 DPA. It contains several genes encoding proteins of interest (Appendix A), including a known function protein (GhCFE1A) that mediates the interplay between the ER network and the actin cytoskeleton (Appendix A; [96]), and other genes encoding actin, villin, profilin, fimbrin, formin, fibrillin, and tubulin [63,64,66,79,82]. In wild *G. barbadense*, it shows very low expression at all timepoints other than 15 DPA. ME17 is related to reactive oxygen species management and is significantly associated with *domestication_development*. It does not display any particularly notable expression patterns, however. Two modules, ME20 and ME53, are related to the microtubule cytoskeleton. They are both significantly associated with all three test conditions. ME20 displays relatively low expression in domesticated *G. barbadense*, with a slight trend of decreasing expression over time and very high expression at 5 and 10 DPA in wild accessions. It contains several genes of interest (Appendix A), encoding proteins such as tubulin, dynamin, kinesin, and microtubule-associated proteins [63,79]. ME53 has higher expression in domesticated *G. barbadense* and higher expression at 10 and 15 DPA in both wild and domesticated accessions. The final relevant module is ME25, related to pectinesterase activity, which plays a role in cell wall extensibility [72]. It is significantly associated with *domestication_development* and shows relatively level expression in domesticated *G. barbadense* with an expression spike at 20 DPA; in wild *G. barbadense* this module has more variable expression.

### 3.5. Homoeolog Module Separation

Because both *G. barbadense* and *G. hirsutum* are allopolyploids, they contain duplicated genes (A- and D-homoeologs) for most genes in the genome. These gene pairs experienced shared ancestry until the divergence of the diploid progenitors of the allopolyploid, after which they may have acquired species-specific differences. In the case of *G. barbadense* and *G. hirsutum*, only one of the diploid progenitors produces spinnable fiber; therefore, it is interesting to ask whether each homoeolog in a given pair displays equivalent and/or parallel responses to speciation and (subsequently) domestication, or alternatively how independent homoeologs are with respect to these evolutionary transitions. To explore this, we examined the modular composition of paired and solo homoeologs in three constructed coexpression networks (i.e., the meta-network of both species, the *G. hirsutum* species network, and the *G. barbadense* species network). In all three coexpression networks, the number of A- and D-homoeologs contained within each network was approximately equivalent, ranging from 30,938 D-homoeologs in the *G. barbadense* network to 31,853 A-homoeologs in the meta-network (Appendix A). Within each network, the difference between the overall number of A- and D-homoeologs included was at most 58, indicating no broad bias in homoeolog usage in these fiber coexpression networks; however, the composition of individual modules within each network exhibited greater variability, ranging from 26 to 90% A-homoeologs (corresponding range for D-homoeolog composition = 10 to 74%). The *G. hirsutum* fiber network constructed here exhibits a slight bias toward D-homoeologs in the composition of most (42 out of 76) modules (Appendix A), whereas most *G. barbadense* (31 of 56) modules are slightly biased towards A-homoeologs (Appendix A). Despite this large variability in module composition and the biases detected, few modules exhibit a significant subgenome bias with respect to homoeolog composition (Appendix A). Of the 42 biased modules in *G. hirsutum*, only four modules (5%) exhibit significant bias: 3 with A-homoeolog biases and 1 with a D-homoeolog bias. Conversely, more *G. barbadense* modules exhibit significant bias (10 modules; 18%); however, these are evenly split as 5 A-biased and 5 D-biased modules. These results characterize the sometimes subtle differences in the fiber coexpression network between species, here with respect to homoeolog usage, and also indirectly support the idea that responses to domestication have been different in the two species.

Because homoeologs represent duplicated genes with recent independent ancestry (for allopolyploids), it is interesting to consider how often homoeolog pairs are placed in the same module (i.e., perform the same or similar functions) versus how frequently they are placed in different modules, potentially reflecting expression and/or functional divergence. While most modules exhibit no significant bias in homoeolog composition, in many cases, this lack of bias obscures the inference of homoeolog expression divergence. That is, the A- and D-homoeologs for the same ancestral gene were often placed in different modules (Table 3; Appendix A) in all three networks, indicating some degree of functional divergence between A- and D-homeologs in terms of coexpression network structure, as previously noted [27]. In the meta-network, approximately equal numbers of homoeologs (31,853 A-homoeologs vs. 31,822 D-homoeologs) were placed into network modules, but only ~37% (23,754 homoeologs, or 11,877 pairs) were placed in the same module; the remaining ~63% were placed in separate modules. Similar statistics were seen in the *G. hirsutum* network, where 31,055 A-homoeologs and 31,032 D-homoeologs were placed into modules, and only about one-third (22,396 homoeologs, or 11,198 pairs) of homoeolog pairs were placed into the same module. Interestingly, the *G. barbadense* network is slightly different: the overall homoeolog composition of the network remains roughly the same (i.e., 30,996 A- and 30,938 D-homoeologs), but here only about 23% of the module is composed of homoeolog pairs (i.e., 14,540 homoeologs, or 7270 pairs; Table 3) potentially suggesting greater expression divergence between homoeologs in *G. barbadense* than in *G. hirsutum*.

### 3.6. Module Correspondence and Preservation

An interesting outcome of the foregoing results is evidence for a somewhat divergent means of arriving at a convergent phenotype; however, the high dimensionality of the data may obscure similarities between the two species. We thus used module correspondence and preservation analyses to determine how well the fiber coexpression network in *G. hirsutum* is reiterated in *G. barbadense*. We employed two complementary statistical methods for evaluating the preservation between these species, medianRank and Zsummary [56], both of which assess the connectivity and density between modules to identify and compare hub genes. Modules with low medianRank scores and/or high Zsummary scores are generally considered better preserved. While there is no established cutoff for medianRank scoring, modules with a high Zsummary score (>10) are considered well-preserved, whereas modules with a Zsummary score between 2 and 10 are considered weakly preserved. In general, modules with a high medianRank and a low Zsummary (<2) are considered not well-preserved [57].

Module correspondence (Figure 5) indicates that modules from each species do not correspond 1:1 with modules in the other species, and many do not exhibit any correspondence between species. Less than half of the modules in *G. hirsutum* (31 modules; 41%) and *G. barbadense* (19 modules, 34%) exhibit statistically significant (Fisher’s exact test; *p* < 0.05) correspondence with at least one module in the other species. Additionally, this correspondence is frequently 1: many; that is, most modules in the *G. hirsutum* network correspond to many modules in the *G. barbadense* network and vice versa, indicating that the modular structure is, overall, not well preserved between species (but see Figure 6 and the following). Likewise, module preservation tests between *G. hirsutum* and *G. barbadense* also suggest general divergence between these fiber coexpression networks. Of the 56 modules tested, the limiting number based on the *G. barbadense* network, most (37, or 66%) exhibit little to no preservation (i.e., Zsummary < 10). Of these, only ten are weakly preserved (2 < Zsummary < 10), whereas 27 (48% of tested modules) exhibit no preservation (Zsummary < 2; Appendix A). These results are generally reiterated by medianRank in that the 19 modules that exhibit strong preservation (Zsummary > 10) also exhibit the lowest medianRank scores. The larger modules (i.e., ME1-M10) generally were among those with high preservation (except ME2 and ME8), although some smaller modules were considered well-preserved.

Overall, 19 modules (34%) were considered well-preserved, many significant for development and/or domestication (Appendix A) in either *G. hirsutum* and/or *G. barbadense*. Of the five best-preserved modules (ME3, ME16, ME20, ME17, and ME43), only ME3 had functional annotations in both species, which included regulation of many cellular processes, including RNA synthesis, macromolecule metabolism, metabolic processes, transcription, and gene expression. Taken together, they show clear module differences between these independently domesticated species (Figure 6), and the general lack of preservation between *G. hirsutum* and *G. barbadense* suggests that the underlying genetic changes resulting in a similar fiber phenotype between *G. hirsutum* and *G. barbadense* have led to extensive changes in network structure both throughout speciation and under domestication. This again highlights the different transcriptomic dimensions of fiber development and domestication in the two species, despite their close relationships.

## 4. Discussion

### 4.1. Independent Domestication Has Uniquely Impacted Two Polyploid Cotton Species

In addition to its agronomic importance, *Gossypium* is also scientifically significant as a model system for studying diploid diversification and allopolyploidization. As reviewed elsewhere [21,97,98], two *Gossypium* diploids, one from Africa-Asia and the other from the Americas, hybridized during the Pleistocene, underwent genome doubling, and diversified into a new allopolyploid clade now represented by seven allotetraploid species, including the two domesticated species studied here [97,98,99,100]. Remarkably and only recently (5000–8000 years ago), in an evolutionary sense, two of the allotetraploid species and two diploid species were domesticated, each independently, in four different regions of the world in two hemispheres [19,20,22,101]. This natural and human-influenced context provides a remarkable opportunity to teach us about the comparative genomic basis of superficially similar morphological transformations that accompanied strong directional human selection under domestication. Here we focus on the two polyploid domesticates, *G. hirsutum* and *G. barbadense*. Their independent domestication provides a unique and powerful opportunity to evaluate the nature of the effects that domestication has had on the genes, biosynthetic pathways, and biological networks in each of these two species over the millennia of geographic diffusion and crop improvement that have occurred since initial domestication from wild progenitors.

### 4.2. Comparative Expression Analysis Supports Independent Domestication Mechanisms

Given the superficial similarity of the ancestral (wild) and descendant (crop plant) fiber phenotypes between species, the question arises as to which of these parallel morphological transformations were caused by similar suites of genetic changes, as reflected in the transcriptome. Our results indicate that while there may be cases of genetic parallelism, for the most part, *G. hirsutum* and *G. barbadense* appear to have arrived at the domesticated cotton fiber phenotype through different transcriptional and genetic alterations. This is evidenced in the principal component analysis by the lack of overlapping clusters related to domestication (Appendix A) and the near absence of overlap in the suites of genes differentially expressed during fiber development in wild vs. domesticated accessions of both species. This latter point is illustrated by the narrow intersection of DE genes inferred to have been altered by domestication in *G. barbadense* and *G. hirsutum*; of the 24,627 and 7608 total cases of DEG in *G. barbadense* and *G. hirsutum* (representing 16,485 and 6630 non-redundant genes, respectively) in the two species, only 2259 were differentially expressed in both species, of which only 297 exhibited DE in the same direction and in the same stage. Also notable is that interspecific expression divergence began before domestication, demonstrated by the existence of two species-specific, developmentally relevant modules (i.e., ME4 and ME5, collectively representing 6663 genes).

*Gossypium hirsutum* has a more dynamic transcriptome than *G. barbadense*, particularly early in fiber development, as indicated, for example, by the substantially higher numbers of DEG (Figure 1). This lower level of differential expression during development in *G. barbadense* may be partially explained by a fundamental difference in the genetic distances of the comparisons in the two species; that is, in *G. hirsutum*, the wild representative is undoubtedly a wild plant, with extant populations scattered in several coastal regions in the Gulf of Mexico. In contrast, in *G. barbadense*, truly wild accessions have not been identified with certainty, and thus what we refer to as “wild” *G. barbadense* may represent a naturalized derivative of a primitively domesticated form that became reestablished, perhaps even thousands of years ago in the small natural range of the species in western Peru and Ecuador [21]. To the extent that this is true, one might expect lower genetic divergence and hence less transcriptomic divergence between wild and domesticated forms of *G. barbadense* than in *G. hirsutum*. As a non-mutually exclusive alternative, it may be that domestication in *G. barbadense* entailed fewer genetic-transcriptomic transformations for any number of biological and historical reasons (e.g., more intense human-mediated selection in *G. hirsutum*).

In addition to this fundamental quantitative difference in the level of transcriptomic change, the patterns of DGE over the course of fiber development also differ; in *G. barbadense*, the number of DGE is similar between 5 and 10 DPA, and 10 and 15 DPA, and increases between 15 and 20 DPA, whereas in *G. hirsutum* DGE is higher early and late (between 5 and 10 DPA, and between 15 and 20 DPA) than it is in the middle of development, between 10 and 15 DPA. We note that higher DGE is expected between 15–20 DPA, as this is the time period in both species where fiber cells enter the transition stage, during which primary cell wall elongation begins to slow, the fiber lays down the winding cell layer, and secondary wall synthesis begins.

When comparing wild and domesticated accessions in both species, generally lower differential gene expression across development was observed in the domesticates (Table 2). One possible explanation for this is that domesticated plants have more canalized developmental programs versus wild accessions, as suggested by the cottonseed transcriptome [58]. With respect to the domesticated cotton fiber, a salient example may be the general repression of the lignin biosynthetic pathway, which results in the deactivation of an entire suite of genes that would otherwise contribute to expression level divergence throughout development [67].

When comparing wild and domesticated accessions of each species at each timepoint, similar patterns of DGE emerge as observed when comparing across timepoints: higher levels of DGE in *G. hirsutum* when compared to *G. barbadense*, higher DGE at 20 DPA in *G. barbadense* (versus *G. hirsutum*), and higher DGE between wild and domesticated *G. hirsutum* at all other stages, peaking at 15 DPA (Table 2). Notably, this mirrors results regarding flowering time neutrality in the two species, where flowering time is controlled by a few loci of major effect in *G. barbadense*, whereas many loci of cumulative effect produce the same phenotype in *G. hirsutum* [102,103]. The observed higher levels of DGE between wild and domesticated *G. barbadense* at 20 DPA, however, are likely due to domesticated *G. barbadense* having a longer elongation period, which can last as late as 25 DPA; consequently, the transcriptional overlap of elongation and secondary wall synthesis occurs for a longer period than it does in *G. hirsutum* or wild *G. barbadense* [67], leading to greater expression differences between wild and domesticated *G. barbadense* at this stage. The peak in *G. hirsutum* DGE at 15 DPA may similarly indicate a transcriptional shift (associated with the transition stage) that is offset to an earlier timepoint in *G. hirsutum*.

When comparing wild *G. hirsutum* to wild *G. barbadense*, there is a striking divergence in expression profiles, indicating that these two species utilize different transcriptional pathways during fiber development, even prior to domestication (Figure 2); however, this conclusion should be tempered until truly wild *G. barbadense* accessions have been studied. Interestingly, when comparing domesticated *G. hirsutum* to domesticated *G. barbadense*, there are even higher levels of differential gene expression between the two species at every timepoint, exceeding the levels seen in all other comparisons discussed here (Table 2). The fact that transcriptional differences are greater between the domesticates than the wild accessions indicates that domestication has increased transcriptional divergence between these two species, even in light of historical introgression and offset in developmental chronology. These expression patterns support the hypothesis that domestication impacted *G. hirsutum* and *G. barbadense* differently, resulting in greater transcriptional divergence post-domestication than post-speciation despite being domesticated for a convergent fiber phenotype. Further study of these and other independent domestication events, such as in *Phaseolus* [104], will enable a deeper understanding of the relative degree of repeatability under directional selection or if, instead, domestication paints with a unique brush every time.

### 4.3. Network Analysis Shows Substantial Differences between G. hirsutum and G. barbadense

Network analysis indicates that the *G. hirsutum* coexpression network consists of 76 coexpression modules, whereas the *G. barbadense* network consists of 56, a notable difference in module number. Interestingly, the *G. barbadense* network is more similar to the meta-network constructed from all samples, which contained 58 modules. Most of the modules in the meta-network had significant associations with domestication, whereas the individual networks only exhibited significant associations with domestication in about half of the modules. The meta-network contained 25 modules that were significant for both species and domestication, 15 modules that were significant for domestication but not species, and ten modules that were significant for species and not domestication. This indicates that domestication likely has a greater impact on the network topology than speciation, as was reflected in the PCA. The meta-network contains modules with significant associations with both domestication and species, indicating that they may be targets of parallel domestication that were altered differently between species. ME7, in particular, provides an excellent example of this; it has high expression exclusively in domesticated *G. hirsutum*, and functional enrichment indicates that it is related to reactive oxygen species management and the fiber cytoskeleton, both of which are key to fiber development [74,76,105]. Similarly, ME14, another module related to the cytoskeleton, is significantly correlated with domestication, and shows higher expression in *G. hirsutum* than in *G. barbadense*.

Examination of the *G. hirsutum* and *G. barbadense* networks reveals that the networks are substantially different beyond simple module assignments. The gene membership of these modules differs between species, with each module in *G. hirsutum* corresponding to multiple modules in *G. barbadense* and vice versa, supporting the module preservation tests indicating that most modules in these species are not well preserved between them. There are also differences on a functional level between these species; the top five modules for each have very different functional enrichment. In *G. hirsutum*, the top five modules have functions that include signaling, regulation of biological processes, DNA, RNA, lipid, and protein metabolism, synthesis, and localization, microtubule cytoskeleton activity, reactive oxygen species management, hormone metabolism, ribosome synthesis, and mitochondrial processes. In *G. barbadense*, the top five modules have functions that include the regulation of cellular processes, DNA synthesis and repair, protein synthesis, localization, folding, polysaccharide metabolism, actin binding, and organic acid biosynthesis. In short, there are drastic differences between the individual species networks, both in terms of module preservation and function, and the meta-network suggests that these differences are due in part to domestication in addition to simple species divergence.

Despite these differences, there are some similarities between the two networks. Many modules with significant associations with domestication perform similar functions in both species. These include modules related to polysaccharide synthesis, the cytoskeleton and intracellular transport, and reactive oxygen species management. These processes are essential for fiber development and have been impacted by domestication. This indicates that while domestication has led these species down unique pathways, it is ultimately leveraging overlapping systems to arrive at a similar phenotype.

In conclusion, although earlier work has shown that domestication alters module membership in the cultivated vs. wild representatives of *G. hirsutum* [58,59], comparisons between the domesticated polyploid cotton species have not been made. Module preservation tests show that nearly half of the network modules are not well preserved between *G. hirsutum* and *G. barbadense*. Despite the selection for convergent fiber phenotypes and introgression in elite cultivars [18], the coexpression networks in these two species are highly dissimilar. This indicates unique but overlapping domestication pathways and speaks to the importance of considering the transcriptome when addressing questions of domestication.

### 4.4. Polyploidy and Homoeolog Responses to Evolutionary Transitions

An interesting dimension to domestication in *G. hirsutum* and *G. barbadense* is the presence of polyploidy-duplicated genes that arise from two species that evolved independently on different continents for approximately 5–10 million years, during which only one developed the spinnable fiber phenotype. This framework leads naturally to questions regarding whether biases exist in homoeolog usage, particularly in the context of domestication. Previous work on cotton has suggested a general D-genome bias in QTLs associated with domestication [26,106,107,108], although surveys of homoeolog expression bias have been less clear [27,109,110]. Notably, these biases appear mostly vertically inherited, although some evidence indicates that homoeolog expression bias also evolves post-polyploidization and during domestication [27,58,109,110,111]. Here we find few changes in homoeolog expression bias under domestication (Appendix A), which is expected given analyses of allelic expression in wild x domesticated *G. hirsutum* hybrids [112]. In an analysis of the influence of cis and trans regulation on homoeolog expression, Bao et al. (2019) found that more than 90% of homoeologs exhibit no significant change in expression under domestication, refuting the expected genome-wide bias toward alleles arising from the fiber-producing parent. This absence of general bias is also reflected in the coexpression network analyses, where the overall modular composition was generally unbiased with respect to homoeolog origin. While these results may generally suggest that homoeologs may retain a certain degree of interchangeability, the frequent placement of A- and D-homoeologs in different modules (Appendix A) indicates some degree of functional divergence may differentiate A- and D-homeologs of the same gene with respect to the fiber coexpression network structure. While this has previously been noted for *G. hirsutum* [27] in the context of previous observations [112], these results indicate that domestication and speciation at the polyploid level have not radically altered homoeolog expression bias. We note, though, that the multiple interacting *cis*- and *trans-*controls on gene expression in a polyploid context [112] likely generate modular connectivities and relationships that are multidimensional and transcriptomically complex, thus potentially generating the distinct modular organizations observed in the two cultivated cotton species.

## 5. Conclusions

Cotton fiber has the distinction of being independently domesticated four times for the same general phenotype in four different species. Underlying these broad, similar phenotypes are smaller but agronomically significant, morphological differences resulting from their independent speciation and domestication paths. The foregoing explores the powerful impact domestication has had on the fiber transcriptome of the two domesticated polyploid cotton species, *G. hirsutum* and *G. barbadense*, resulting in remarkably divergent transcriptomes and impacting expression divergence more in the 5000–6000 years since domestication than the prior half million years or so. Despite the substantial divergence in the organization of the transcriptome, the presence of conserved modules and module functions in the domesticates highlights potential parallel targets of selection. Future fine-scale analyses between these domesticated species and their wild progenitors will be paramount in understanding the origin and consequences of this remarkable expression divergence that underpins a generally convergent phenotype.

## Figures and Tables

**Figure 1 genes-14-01301-f001:**
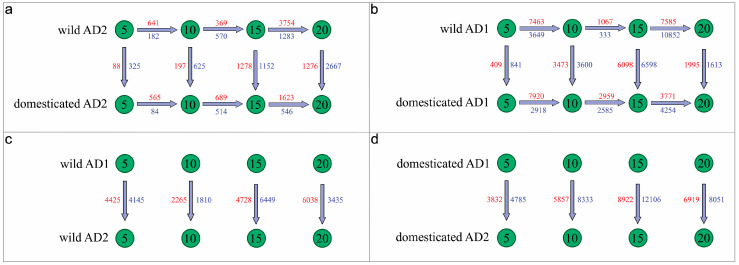
Differential gene expression between fiber samples of *G. barbadense* (AD_2_) and *G. hirsutum* (AD_1_). Comparisons of four developmental timepoints (5, 10, 15, and 20 days post-anthesis) were conducted (**a**) within AD_2_, (**b**) within AD_1_, and between (**c**) wild AD_1_ *versus* wild AD_2_, and (**d**) domesticated AD_1_ *versus* domesticated AD_2_. Arrows indicate the direction of comparison from test to reference. Red indicates genes that are more highly expressed. Blue indicates genes that are lowlier expressed.

**Figure 2 genes-14-01301-f002:**
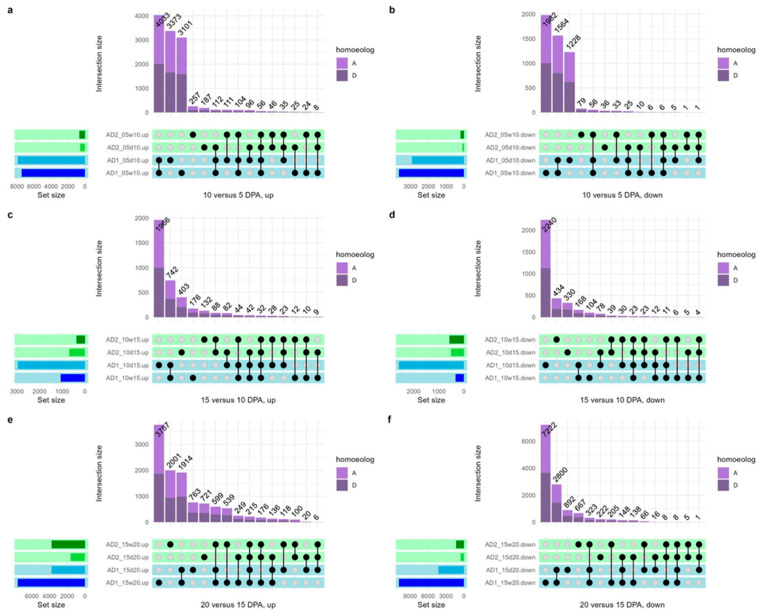
Upset plots comparing DGE overlaps in *G. hirsutum* (AD_1_) and *G. barbadense* (AD_2_) wild and domesticated accessions between developmental timepoints. (**a**) Upregulated genes at 10 vs. 5 days post anthesis (DPA); (**b**) Downregulated genes at 10 vs. 5 DPA; (**c**) Upregulated genes at 15 vs. 10 DPA; (**d**) Downregulated genes at 15 vs. 10 DPA; (**e**) Upregulated genes at 20 vs. 15 DPA; and (**f**) Downregulated genes at 20 vs. 15 DPA. Set sizes (i.e., the number of genes in each category) are displayed on the left side of each plot, and the intersections are displayed above. The total number of genes in each intersection category (denoted by the black connected dots) is listed above the bar, and each bar is colored based on the homoeolog composition (A vs. D) of that interaction category. Sets are named based on which species are involved (i.e., “AD1” or “AD2”), the DPA(s) under consideration (i.e., 5, 10, 15, and/or 20), the condition (i.e., w = wild, d = domesticated) and whether the set represents up- or downregulated genes. In all instances, the latter DPA is contrasted with the earlier DPA.

**Figure 3 genes-14-01301-f003:**
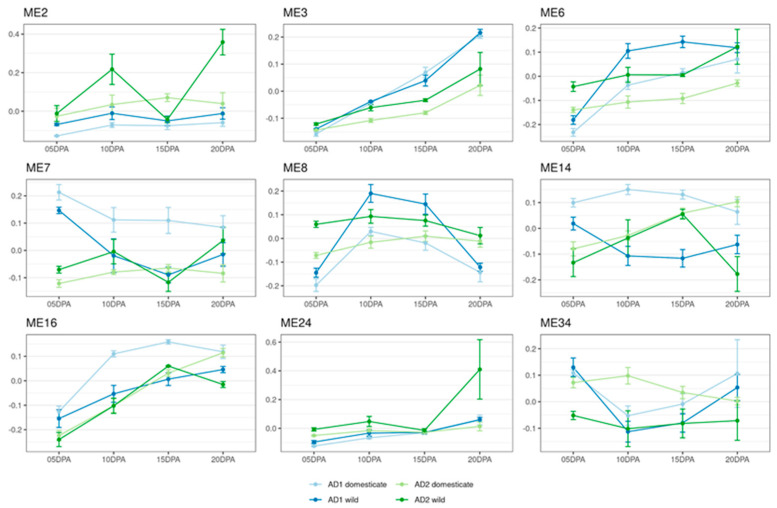
Eigengene expression patterns for nine modules of interest in the meta-coexpression network. Expression patterns are given for *G. hirsutum* (AD1, blue) and *G. barbadense* (AD2, green) for both domesticated (light) and wild (dark) accessions. Numbers on the *x*-axis labels denote the timepoints for the sample. Error bars indicate standard error. Module numbers are derived from WGCNA; modules with higher numbers are composed of fewer genes. Significance with respect to domestication and/or development is given in Appendix A.

**Figure 4 genes-14-01301-f004:**
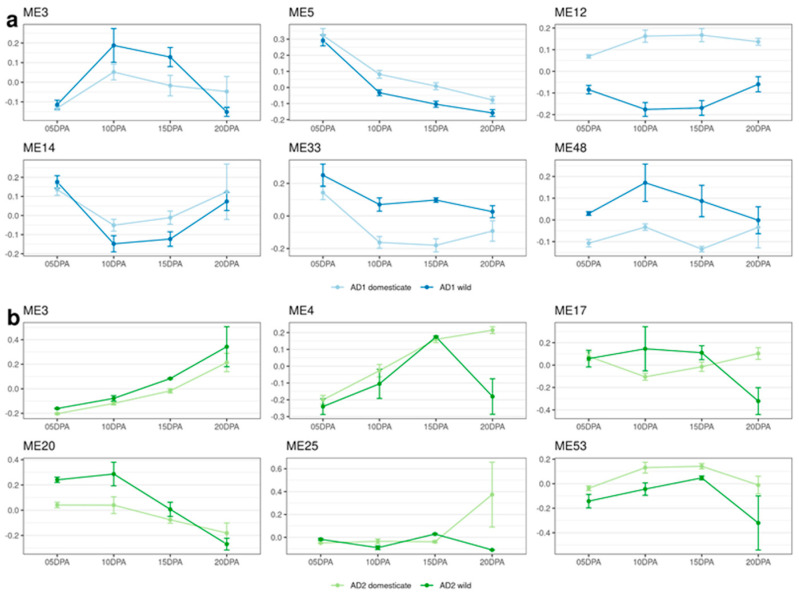
Eigengene expression patterns for modules of interest in (**a**) the *G. hirsutum* (AD_1_) species network and (**b**) the *G. barbadense* (AD_2_) species coexpression network. Expression patterns are given for both wild (dark) and domesticated (light) accessions. Numbers on the *x*-axis labels denote the timepoints for the sample. Error bars indicate standard error. Module numbers are derived from WGCNA; modules with higher numbers are composed of fewer genes. Significance with respect to domestication and/or development is given in Appendix A.

**Figure 5 genes-14-01301-f005:**
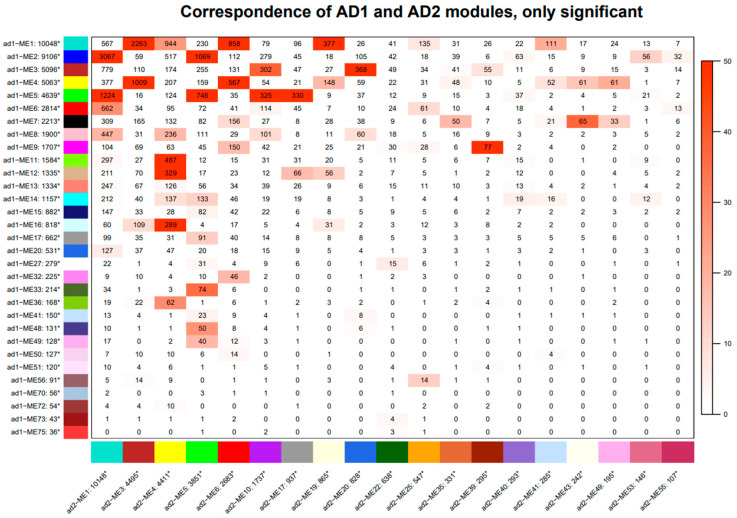
Correspondence of significant modules in the *G. hirsutum* (AD_1_, vertical axis) and *G. barbadense* (AD_2_, horizontal axis) networks. Numbers in each cell indicate the number of genes shared between the corresponding AD_1_ (row) and AD_2_ (column) modules. Cell shading denotes the significance of correspondence based on Fisher’s exact test, with darker coloring indicating higher levels of correspondence, as measured by −log10 (*p-*value).

**Figure 6 genes-14-01301-f006:**
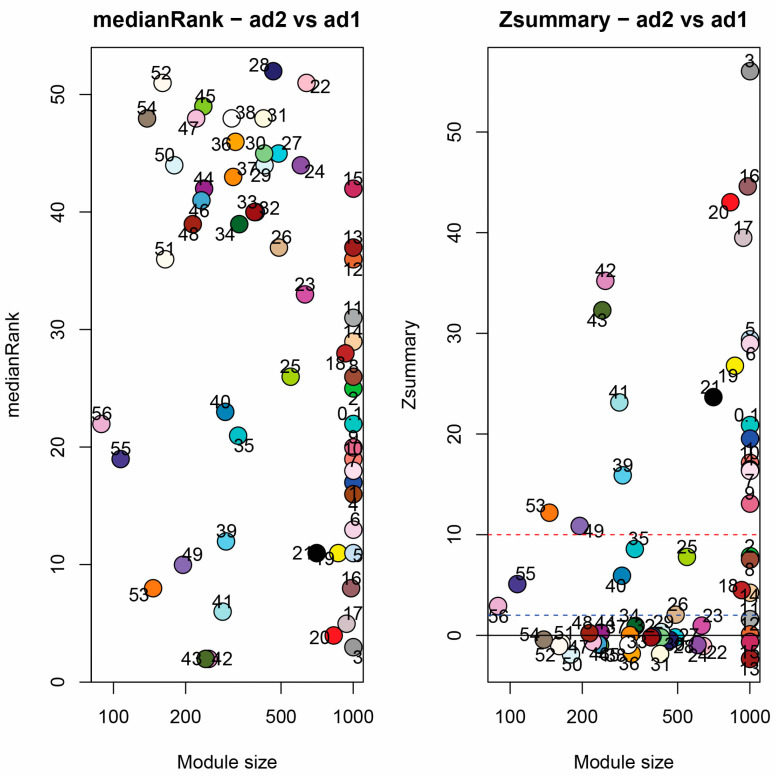
Two tests of module preservation. On the left, medianRank, indicating the preservation of *G. barbadense* (AD2) modules compared to *G. hirsutum* (AD1) modules. Low medianRank scores indicate high preservation. On the right, ZSummary, indicates the preservation of *G. barbadense* (AD2) modules compared to *G. hirsutum* (AD1) modules. High ZSummary scores indicate high preservation. Modules below the red dashed line at 10 are considered not well preserved.

**Table 1 genes-14-01301-t001:** Accessions sampled. Accessions sharing the same [species + domestication] status were used as biological replicates.

Species	Accession	Domestication Status
*Gossypium hirsutum*	CRB252	domesticated
	Maxxa	domesticated
	TM1	domesticated
	TX665	wild
	TX2094	wild
	TX2095	wild
*Gossypium barbadense*	Pima S6	domesticated
	Pima S7	domesticated
	Phy76	domesticated
	GB0303	wild
	GPS52	wild
	K101	wild

**Table 2 genes-14-01301-t002:** DGE gene number results. Differential expression comparisons for *G. barbadense* and *G. hirsutum*, between wild and domesticated or among DPA. Columns indicate the number of genes differentially expressed at *p-*adjusted < 0.05. Numbers in parentheses indicate the number of genes up and downregulated for that comparison (up: down).

Material	DPA	*G. barbadense*	*G. hirsutum*	Interspecies	DPA
wild	10 vs. 05	823 (641:182)	11,112 (7463:3649)	8570 (4425:4145)	5
	15 vs. 10	939 (369:570)	1400 (1067:333)	4075 (2265:1810)	10
	20 vs. 15	5037 (3754:1283)	18,437 (7585:10852)	11,177 (4728:6449)	15
				9473 (6038:3435)	20
domesticated	10 vs. 05	649 (565:84)	10,838 (7920:2918)	8617 (3832:4785)	5
	15 vs. 10	1203 (689:514)	5544 (2959:2585)	14,190 (5857:8333)	10
	20 vs. 15	2169 (1623:546)	8025 (3771:4254)	21,028 (8922:12106)	15
				14,970 (6919:8051)	20
wild versus domesticated	5	413 (88:325)	1250 (409:841)		
	10	822 (197:625)	7073 (3473:3600)		
	15	2430 (1278:1152)	12,696 (6098:6598)		
	20	3943 (1276:2667)	3608 (1995:1613)		

**Table 3 genes-14-01301-t003:** The modular composition of A- and D-homoeologs in *G hirsutum* (AD_1_) and *G. barbadense* (AD_2_). Solo homoeologs are placed in a module, while the other homoeolog is placed in a different module. A- and D-dominant modules are those modules that are significantly biased toward the A- or D-subgenome.

	Meta	AD1	AD2	AD1-AD2 Consensus
**Total module genes**	**63,675**	**62,084**	**61,934**	**57,019**
Solo A-homoeolog	19,976 (31.4%)	19,857 (32.0%)	23,726 (38.3%)	22,171 (38.9%)
Solo D-homoeolog	19,945 (31.3%)	19,834 (31.9%)	23,668 (38.2%)	22,210 (39.0%)
Homoeolog pairs	11,877 (37.3%)	11,198 (36.1%)	7270 (23.5%)	6319 (22.2%)
**Total modules**	**58**	**76**	**56**	**165**
A-dominant module	7	3	5	5
D-dominant module	5	1	5	6

## Data Availability

All code is available at https://github.com/Wendellab/AD2-vs-AD1 (accessed on 19 June 2023). Sequence data is deposited under PRJNA TBD.

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
