# Peer review of "Domestication over Speciation in Allopolyploid Cotton Species: A Stronger Transcriptomic Pull"

_genes, 2023, doi:10.3390/genes14061301_

Round 1

Reviewer 1 Report

With the approach in the evolutionary history of two Gossypium species domestication, the manuscript analyzed the co-expression in several transcriptomics data. To answer different questions about the domestication aspect, a large amount of the data were analyzed and some interesting insights were obtained. However, the text and figures are sometimes confuse, showing difficulty in describing the relevant network. In general, I suggest the authors relooked the manuscript, and take out to excess information common in the omes approaches that make the text much described.

Along the work, AMOVA tests were performed to identify the association between the genes in Meta-coexpression network analysis, right? Did the authors perform the statistical analysis inside the modules? The difference statistical should be sinalized in the figure 3 and 4 to better support the results discurssion. Points without sigificant difference not should be discussed.

The text in “Module correspondence and preservation” topic is confusing.

Minor points:

Line 114: Replace “4)” with “ Four” to  keep the standard in the text

Figure 1: The figure should be cited first in the text. I suggest changing the “a” square to “b” keeping the same format of the legend title.

Figure 2 is not so clear. What represents the black connected dots? What exactly refers “set size”? The legend could be improved.

Figure 3 and 4:  the meaning of the Module numbers should be explicit in the legends.

Check the Table 3 formation. It is different from table 2 and 3

Author Response

We thank the reviewer for their time and effort in reviewing our manuscript. We have addressed the concerns to the best of our ability, as outlined below.

Enumerated comments:

  1. However, the text and figures are sometimes confuse, showing difficulty in describing the relevant network. In general, I suggest the authors relooked the manuscript, and take out to excess information common in the omes approaches that make the text much described.

We appreciate the author’s desire for a less dense manuscript; however, we do not find any specific redundancy or irrelevance. We hope that the other modifications have served to clarify the manuscript with respect to this comment.

  1. Along the work, AMOVA tests were performed to identify the association between the genes in Meta-coexpression network analysis, right? Did the authors perform the statistical analysis inside the modules? The difference statistical should be sinalized in the figure 3 and 4 to better support the results discurssion. Points without sigificant difference not should be discussed.

We have added a statement regarding significance to the figure legends of both and refer the reader to the appropriate (existing) table (S9) to understand if each module is significant with respect to domestication and/or development

  1. The text in “Module correspondence and preservation” topic is confusing.

We have reviewed the text, but are uncertain as to what specifically the reviewer finds confusing. It is a complicated topic, so we realize that it will be a slower section to read. 

  1. Line 114: Replace “4)” with “ Four” to  keep the standard in the text

We thank the reviewer for catching this editing error. We have made the suggested change.

  1. Figure 1: The figure should be cited first in the text. I suggest changing the “a” square to “b” keeping the same format of the legend title.

We thank the reviewer for this comment. Figure 1 is cited before all other figures, so the final placement will be decided by the production team. As for matching the legend title to the order of the figure, this is a good suggestion. We have changed the legend title to match the order of the figure.

  1. Figure 2 is not so clear. What represents the black connected dots? What exactly refers “set size”? The legend could be improved.

We added some text to the legend to guide those people that are not familiar with upset plots.

  1. Figure 3 and 4:  the meaning of the Module numbers should be explicit in the legends.

We added the following text to the manuscript: Module numbers are derived from WGCNA; modules with higher numbers are composed of fewer genes.

  1. Check the Table 3 formation. It is different from table 2 and 3

We have checked these tables, but are unsure what the reviewer would like to see changed. 

Reviewer 2 Report

1.     Is the Title appropriate to describe whole story of the research? Can be Improved with more detailed and specific information related to the research.

2.     Does the Abstract represent the research? Can be Improved by adding more background information why did you do comparative studies and I think it will be better if you add the application of this research in fiber industry.

3.     Do the authors summarize the main research question and key findings? No, please add the main research question and key findings of your research.

4.     In Introduction, Is the content succinctly described and contextualized with respect to previous and present theoretical background and empirical research (if applicable) on the topic? Can be improved by adding more information about the previous studies related to the two cotton species.

5.     Do the authors identify other literature on the topic and explain how the study relates to this previously published research? Can be improved by making smooth connection among the information of the previous studies and your research in Introduction section. 

6.     Is the Objectives of the research address correctly? Can be improved by using more powerful words.

7.     Is the main question addressed by the research? Yes.

8.     How original is the topic? The topic is original but I saw many studies with similar topic.

9.     Does the Methodology describe well? Yes.

10.  Is the rationale for the proposed study clear and valid? Yes

11.  Is the methodology technically sound? Will it effectively achieve its aims, and test the stated hypotheses? Yes

12.  Is the methodology feasible and detailed enough to make the work replicable? Yes.

13.  Is the methodology and any analysis made correct and properly conducted? Yes.

14.  Are the experiments or interventions appropriate for addressing the research question? Yes.

15.  Is there enough data to draw a conclusion? Yes.

16.  Do the authors address any possible limitations of the research? No. Please add the limitation of the research and how it will affect your results.

17.  Was data collected and interpreted accurately? Yes.

18.  Do the authors follow best practices for reporting? Yes.

19.  Does the study conform to ethical guidelines? Yes.

20.  Could another researcher reproduce the study with the same methods? In other words, have the authors provided enough information to validate the study? Yes.

21.  Is the statistical analysis adequate? Yes.

22.  Is the Result display in the correct way? Yes.

23.  Are the figures and tables clear and readable? Can be improved with higher resolution.

24.  Are the figure and table captions complete and accurate? Yes.

25.  Are the axes labeled correctly? Yes.

26.  Is the presentation appropriate for the type of data being presented? Yes.

27.  Do the figures and tables support the findings? Yes.

28.  Do the data provide enough evidence for the authors’ conclusions? Please add Conclusion section.

29.  Have the authors provided a sufficient amount of data and information for other researchers to recreate the analyses? Yes.

30.  Does the Discussion describe all of the results? Yes

31.  Are the arguments and discussion of findings coherent, balanced and compelling? Yes.

32.  Is the paper well written? Yes.

33.  Is the text clear and easy to read? Yes. 

34.  Are the conclusions consistent with the evidence and arguments presented? Please add Conclusion section.

35.  Are the conclusions address the main question posed? Please add Conclusion section.

36.  Are the conclusions supported by the data, and do they address the hypothesis? Please add Conclusion section.

37.  Do the results support the conclusions? Please add Conclusion section.

38.  Does the Reference cite appropriately? Can be improved by adding more newest references.

39.  Are relevant data, citations, or references present? Yes.

Please double check the Grammar error.

Author Response

We thank the reviewer for their time and effort in reviewing our manuscript. We have enumerated our response below.

  1.     Is the Title appropriate to describe whole story of the research? Can be Improved with more detailed and specific information related to the research.

We agree with the reviewer, and have retitled the manuscript: Domestication over speciation in allopolyploid cotton species: a stronger transcriptomic pull

  1.     Does the Abstract represent the research? Can be Improved by adding more background information why did you do comparative studies and I think it will be better if you add the application of this research in fiber industry.

Regarding why we did comparative studies, we point the reviewer to the existing text “Comparative analysis of the transcriptome during cotton fiber development in wild and cultivated materials holds promise for revealing how independent domestications led to the superficially similar modern cotton fiber phenotype in upland (G. hirsutum) and Pima (G. barbadense) cotton cultivars.” We hesitate to add additional text here, which will unduly lengthen the abstract.

  1.     Do the authors summarize the main research question and key findings? No, please add the main research question and key findings of your research.

Although not formatted as a question, we have added our main research question to the manuscript: Here we examined the fiber transcriptomes of both wild and domesticated G. hirsutum and G. barbadense to compare the effects of speciation versus domestication…

Regarding the key findings, the reviewer can find those in the abstract on lines 25-31. 

  1.     In Introduction, Is the content succinctly described and contextualized with respect to previous and present theoretical background and empirical research (if applicable) on the topic? Can be improved by adding more information about the previous studies related to the two cotton species.

We have added a couple sentences to the introduction with relevant citations; however, we note that we also contextualize the data within the results themselves. 

  1.     Do the authors identify other literature on the topic and explain how the study relates to this previously published research? Can be improved by making smooth connection among the information of the previous studies and your research in Introduction section. 

We added a couple sentences to the introduction to note the previous research that this is related to.

  1.     Is the Objectives of the research address correctly? Can be improved by using more powerful words.

We respect the reviewer’s opinion here; however, we prefer not to overstate the extent of what the present can address. We do note that these species show “extraordinary [transcriptomic] divergence”, and we hope this will suffice. 

  1.     How original is the topic? The topic is original but I saw many studies with similar topic.

We agree that cotton coexpression network analysis is becoming more common; however, we note that our novelty lies in the design of the experiment -- interspecific and inter-condition for the two most agronomically important species.

  1. Do the authors address any possible limitations of the research? No. Please add the limitation of the research and how it will affect your results.

Perhaps the reviewer missed the caveats on lines 640-653 regarding effects of the “wild” G. barbadense representative and reiterated on lines 688-690? We also mention the possibility of developmental offset on lines 678-586. We have added a sentence to the end of the conclusions that suggest that fine-scale analyses will provide better information regarding how differences in expression have led to the differences in the cotton fiber phenotype that characterize these two species.

  1. Are the figures and tables clear and readable? Can be improved with higher resolution.

We agree and will provide the journal with the original, high resolution images.

  1. Do the data provide enough evidence for the authors’ conclusions? Please add Conclusion section.

Added. 

  1. Are the conclusions consistent with the evidence and arguments presented? Please add Conclusion section.

Added.

  1. Are the conclusions address the main question posed? Please add Conclusion section.

Added.

  1. Are the conclusions supported by the data, and do they address the hypothesis? Please add Conclusion section.

Added.

  1. Do the results support the conclusions? Please add Conclusion section.

Added.

  1. Does the Reference cite appropriately? Can be improved by adding more newest references.

We have added twelve references to the manuscript to highlight similar research.

Comments on the Quality of English Language: Please double check the Grammar error.

We have reviewed the language and find few errors. If the reviewer finds any that linger, we would appreciate the line number so that we can address those specifically.